# Quantitative analysis of seven plant hormones in *Lotus japonicus* using standard addition method

**Takuyu Hashiguchi, Masatsugu Hashiguchi, Hidenori Tanaka, Koki Fukushima, Takahiro Gondo, Ryo Akashi**  *

Department of Animal and Grassland Sciences, University of Miyazaki, Miyazaki, Japan

* rakashi@cc.miyazaki-u.ac.jp

**Data Availability Statement:** All relevant data are within the paper and its Supporting Information files.

## Abstract

Plant hormones have been identified to be versatile signaling molecules essential for plant growth, development, and stress response. Their content levels vary depending on the species, and they also change in response to any external stimuli. Thus, simultaneous quantification of multiple plant hormones is required to understand plant physiology. Sensitive and quantitative analysis using liquid chromatography-linked mass spectrometry (LC-MS/MS) has been used in detecting plant hormones; however, quantification without stable isotopes is yet to be established. In this study, we quantified seven representative plant hormones of *Lotus japonicus*, which is a model legume for standard addition method. Accurate masses for monoisotopic ions of seven phytohormones were determined for high-resolution mass spectrometry (HR-MS). Selected ion monitoring (SIM) mode based on accurate masses was used in detecting phytohormones in the roots, stems, and leaves. Evaluation of matrix effects showed ion suppression ranging from 10.2% to 87.3%. Both stable isotope dilution and standard addition methods were able to detect plant hormones in the roots, stems, and leaves, with no significant differences in using both approaches and thus a standard addition method can be used to quantify phytohormones in *L. japonicus*. The method will be effective, especially when stable isotopes are not available to correct for matrix effects.

## Introduction

Plant hormones are identified as essential small molecules implicated in a variety of fundamental biological processes, including growth, development, and stress response [1]. These hormones are classified into nine groups: auxins, cytokinins, gibberellins, ethylene, abscisic acid, salicylic acid, jasmonic acids, strigolactones, and brassinosteroids. Each class shows bioactivities alone and in combination with other hormones, which is generally known as hormonal crosstalk [2]. Therefore, quantifying multiple plant hormones simultaneously is required to understand plant physiology. Detecting trace amounts of phytohormones in plant tissues employs two major analytical tools, that is, gas chromatography coupled with mass spectrometry (GC-MS) [3] and liquid chromatography coupled to electrospray tandem mass

**Funding:** This work was supported by the National BioResource Project (NBRP) of the Japan Agency for Medical Research and Development (AMED). The funder had no role in study design, data collection and analysis, decision to publish, or preparation of the manuscript.

**Competing interests:** The authors have declared that no competing interests exist.

spectrometry (LC-ESI-MS/MS) [4], methods which have been used over the last decade. LC-E-SI-MS/MS-based quantitative analysis can elucidate plant hormonomics, taking advantage of high selectivity, sensitivity, and specificity for target compounds [5–8]. Unlike GC-MS, electrospray ionization (ESI) is utilized in LC-MS/MS for ionizing analytes of a wide range of polarities without derivatization; however, this method is limited by matrix effects, such as ion enhancement or ion suppression by co-eluting compounds that interfere with target analyte quantification [9]. Several approaches have been taken to date [10–12] in an attempt to mitigate effects and ensure accurate quantitation. First is by reducing the matrix via partial purification, dilution of samples, and injection of small volumes of sample [13]. Second is by changing the ionization mode from ESI to atmospheric pressure chemical ionization (APCI) [14]. Third is by calibrating the matrix effects by applying stable isotope dilution, matrix matching, or standard addition. Plant hormones are often quantified using stable isotope dilution, which involves the addition of stable isotope-labelled counterparts to target analytes [5–7,15,16]. Stable isotope-labelled hormones have substantially the same chemical properties as target compounds. Matrix effects are considered identical with and without labelling, allowing accurate calibration of matrix effects. However, stable isotope labelling is deemed expensive and sometimes unavailable for minor phytohormone metabolites. Matrix matching is also used to calibrate matrix effects; it is often applied to detect drugs and agricultural chemicals [17–19]. This technique can be used without stable isotopes, but it requires a sample matrix without target analyte and is thus not applicable to endogenous compounds in tissues. Another method is standard addition that can be applicable, in theory, to all compounds [12,20]. Standard addition uses actual samples to create individual calibration plots. An analyte is present in both the calibration standards and sample, allowing correction of the matrix effect without stable isotopes. Standard addition is laborious in terms of preparing the calibration standards by sample, but it is still promising especially when stable isotope-labelled phytohormones are unavailable. Nevertheless, there are no reports demonstrating its ability in detecting plant hormones.

In this study, we have validated the simultaneous quantification of seven major plant hormones by standard addition using high-resolution mass spectrometry (HR-MS). We focused on *Lotus japonicus*, a model legume, and quantified plant hormones in its roots, stems, and leaves. The matrix effects were then examined by comparing standards in solvent with standards in matrix. This method was also compared with stable isotope addition. A detailed protocol was developed and is discussed.

## Methods and materials

### Materials

MG20, an experimental strain of *Lotus japonicus*, was obtained from LegumeBase in the National BioResource Project (https://www.legumebase.brc.miyazaki-u.ac.jp/). Isotopically labelled internal standards including [$^2$H$_2$]-gibberellin A$_4$ (GA$_4$), [$^2$H$_6$]-(+)-*cis,trans*-abscisic acid (ABA), [$^2$H$_3$]-brassinolide (BL), [$^2$H$_4$]-salicylic acid (SA), [$^{15}$N$_4$]-*trans*-zeatin (tZ), [$^{15}$N$_4$]-cis-zeatin (cZ), [$^2$H$_6$]-(±)-jasmonic acid (JA), and [$^2$H$_5$]-indole-3-acetic acid (IAA) were purchased from OlChemIm Ltd. (Olomouc, Czech Republic). Gibberellin A$_4$ (GA$_4$) was purchased from Santa Cruz Biotechnology, Inc. (CA, USA). Abscisic acid (ABA), *trans*-zeatin (tZ), and jasmonic acid (JA) were obtained from Tokyo Kasei Kogyo Co. (Tokyo, Japan). Salicylic acid (SA) and indole-3-acetic acid (IAA) were purchased from Wako Pure Chemical Industries Ltd. (Osaka, Japan). Brassinolide (BL) was from Cayman Chemical (MI, USA). *cis*-Zeatin (cZ) was from Santa Cruz Biotechnology Inc. Quartz sand was obtained from Tochu Co. (Aichi, Japan). All the other chemicals were the highest-grade commercially available products.

## Growth condition

*Lotus japonicus* seeds were scarified, subjected to water absorption for 30 min, and sown in quartz sand. The seeds were then germinated at 25˚C and grown in a plant growth chamber with daily cycle of 16 hours of light at 25˚C and 8 hours of dark at 23˚C (BioTRON; Nippon Medical & Chemical Instruments Co., Ltd.). The plants were fertilized with 1000-fold diluted Hyponex® solution (N:P:K = 6:10:5) once a week. One-month-old *L. japonicus* were used in extracting plant hormones.

## Sample preparation

Extraction for plant hormones has been conducted utilizing a previously established protocol with minor modifications [6]. Briefly, leaves, roots, and stems were individually homogenized in liquid nitrogen using a TissueLyser II (Thermo Fisher Scientific). In total, 50 mg of the material was extracted with 1 ml cold 50% acetonitrile. The extract was then purified on a non-selective reversed-phase solid-phase extraction (RP-SPE) using an Oasis HLB cartridge (Waters). The column was activated with 100% methanol and ultrapure water, followed by equilibration with 50% acetonitrile. The sample was loaded onto the cartridge and flow-through collected. The residues of the target hormones were then eluted with 1 ml of 30% acetonitrile. The flow-through and eluted fractions were mixed and were evaporated to dryness in vacuum concentrator for 3 hours. Dried residuals were dissolved in 100 µl of 30% acetonitrile. The protocol was deposited in protocols.io (http://dx.doi.org/10.17504/protocols.io.bqy6mxze).

## Plant hormone determination

Standard addition and stable isotope dilution methods were used in quantifying plant hormones [20–22]. Stable isotope-labelled plant hormones were fortified with samples as well as absolute standard solutions at concentrations ranging from 1.0 ng/ml to 250.0 ng/ml. Neat standard solutions were post-spiked for standard additions with actual sample in order to construct matrix calibration curves in the range of 1.0 ng/ml to 250.0 ng/ml to equalize matrix effects among samples (S1 Fig). Accordingly, actual samples were diluted (dilution factor 1.05). Calibration curves for each phytohormone were constructed for each analyte using the same matrix. Matrix effects (ME = A—B/A*100) were then calculated using the peak areas of A and B, with A identified as a peak area of an analyte in a standard solution and B as a peak area of an analyte in a matrix [9]. A peak area of an analyte derived from the sample was subtracted from B by analyzing the sample beforehand. Recovery rate was calculated by comparing the peak area of each phytohormone present in the sample spiked before SPE and the sample spiked after SPE. Limits of detection (LOD) and limit of quantification (LOQ) were defined as a signal-to-noise ratio of 3:1 and 10:1, respectively.

## Ultra-high performance liquid chromatography linked with high-resolution mass spectrometry (LC-MS/MS)

Purified extracts were separated on a 2.6 µm Accucore C18 LC column (150 mm × 2.1 mm) (Thermo Fisher Scientific) using a linear methanol gradient of 1–100% for 10 min at a flow rate of 0.5 ml/min and a column oven of 40˚C. MS data were acquired in targeted selected ion monitoring (t-SIM) mode using an electrospray ionization Orbitrap Q-Exactive (Thermo Fisher Scientific) linked to an UltiMate 3000 RSLC (Thermo Fisher Scientific). Mass spectrometric conditions were as follows: polarity, positive and negative ionization modes; spray voltage for positive, 3.5 kV; spray voltage for negative, 2.0 kV; sheath gas flow rate, 50; auxiliary gas, 10; sweep gas, 0; heated capillary temperature, 380˚C; S-lens RF level, 50; and auxiliary gas

heater temperature, 350°C. The resolution was then set at 70,000. The AGC target was 5E4. The maximum ion injection time was 200 ms. The isolation window was 10 m/z and offset two to monitor stable isotope-labelled plant hormones.

### Data analysis

Raw data files were analyzed using Qual Browser software in Xcalibur (Thermo Fisher Scientific). For quantification, Quan browser software in Xcalibur 4.2.47 (Thermo Fisher Scientific) was also used. Student's t-test was performed using the Excel software.

## Results

### Plant hormone detection by selected ion monitoring mode (SIM)

Major phytohormones are identified as follows: indole acetic acid (IAA), *trans/cis*-zeatin (tZ and cZ), abscisic acid (ABA), salicylic acid (SA), gibberellin $A_3$ ($GA_3$), jasmonic acid (JA), and brassinolide (BL) [1] (Fig 1).

Stable isotope-labelled gibberellin $A_3$ was commercially unavailable; thus, gibberellin $A_4$ was used instead. An accurate mass of the monoisotopic ion of each plant hormone was first determined by direct infusion in positive and negative ESI mode (S1 Table). Five phytohormones were detected in negative mode, and IAA and BL were only detected in positive mode. Seven phytohormones were injected onto a C18 column, separated, and then detected in the targeted selected ion monitoring (SIM) mode using a measured accurate mass of monoisotopic ion. Except for zeatin, the other six phytohormones showed a distinct retention time and were well separated (S2 Fig). *trans*-Zeatin (tZ) and *cis*-zeatin (cZ) stereoisomers showed close retention times and could not be fully separated in the gradient used. Stable isotope-labelled phytohormones were also analyzed in order to determine accurate mass and retention time; they were then compared to non-labelled hormones (S1 Table and S3 Fig). The retention times were almost the same between the stable isotope-labelled hormones and its corresponding non-labelled counterparts. Matrix effects appear to be of the same extent regardless of labelling. Thus, we established analytical conditions for plant hormone quantification by SIM and the chromatographic patterns suggested the possibility of close correspondence between standard addition and stable isotope dilution methods.

Plant hormone extraction has been basically referred to a previous report [6] with minor modifications and workflow as described in Fig 2.

We spiked moderate levels of seven phytohormone mixtures before and after solid-phase extraction (SPE) in order to examine recovery rates and peak areas for each hormone in the leaves, roots, and stems of *Lotus japonicus* (S2 Table). IAA and BL were determined to show slightly lower recovery rates, but other recovery rates were around 80–100%, indicating that almost all phytohormones can be recovered from the tissues of *Lotus japonicus* by this extraction.

### Non-negligible matrix effects of root, stem, and leaf from *Lotus japonicus* in plant hormone determination

Matrix effects can often cause difficulties in detecting target analytes by reducing or increasing the sensitivity of quantification [12]. Thus, we examined the matrix effects for the quantification of plant hormones in the tissue extracts from *L. japonicus* (Table 1).

All tissues showed matrix effects ranging from 10.2% to 87.3%, and all matrix effects were more than 0, indicating ion suppression. cZ also showed the highest ion suppression in root extracts. However, the lowest ion suppression was found in BL in root extracts. Ion suppression might decrease the peak area of phytohormones in the presence of matrix, thereby

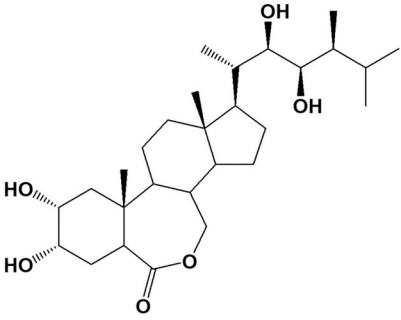

**Fig 1. The chemical structures of the seven plant hormones used in this study.**

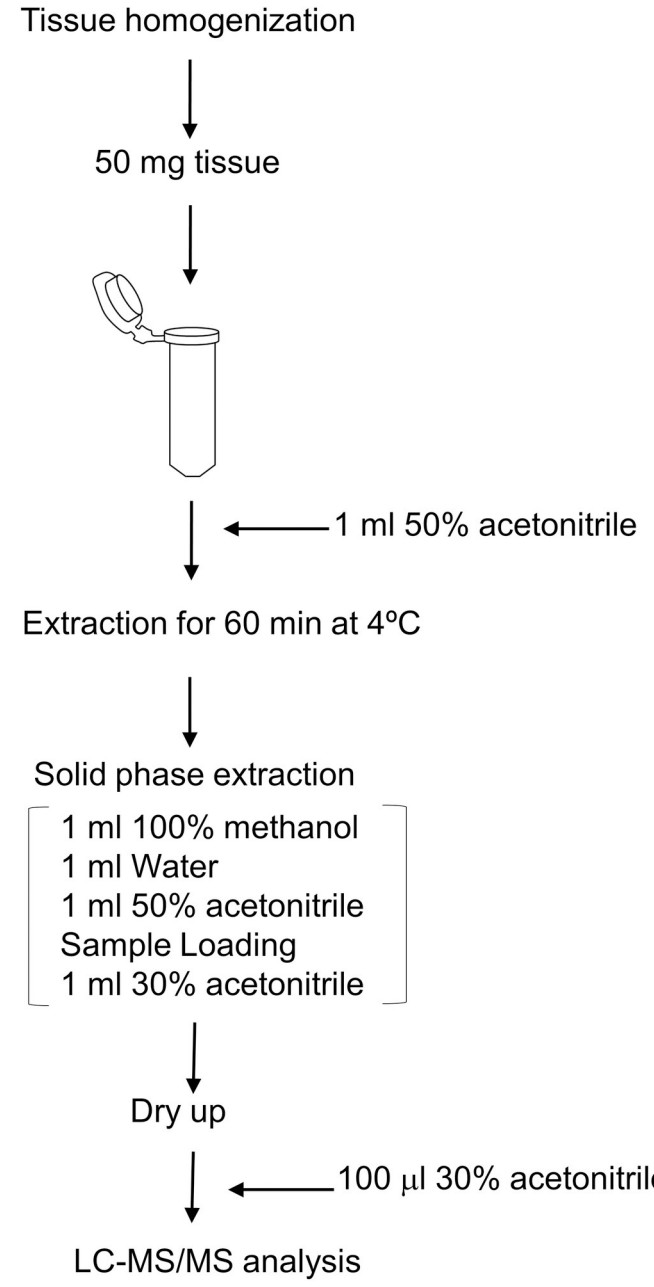

**Fig 2. Experimental workflow for extraction of phytohormones.** Fifty mg of tissues was extracted with 50% acetonitrile; it was then partially purified by RP-SPE. Dried residue was dissolved in 30% acetonitrile and was used for LC-MS/MS analysis.

underestimating target analyte content when an absolute standard curve is used in quantification. Thus, mitigating the matrix effect, such as using stable isotope dilution, is indispensable for the accurate quantification of plant hormones in *Lotus japonicus*.

## Quantification of plant hormones by stable isotope dilution

Proper calibration has been identified to be essential for obtaining reliable results for targeted compounds. Generally, an internal standard calibration method using stable isotope-labelled

target compounds is adopted. We then extracted the plant hormones to quantify them in the tissues of *L. japonicus*. Extracts were mixed with the corresponding stable isotope-labelled phytohormones and used in LC-MS/MS analysis. An absolute standard curve corrected by stable isotope addition was constructed, and plant hormones were then quantified (Table 2).

tZ was deemed unquantifiable because a large peak appeared before the target peak. The other seven hormones were quantified. SA was determined to be the most abundant hormone found in the leaves of *L. japonicus* (2397.5 ± 203.5 (pmol/g FW)). JA, ABA, IAA, and BL were also detected in an order of decreasing concentration. Conversely, root extracts contained the highest JA concentration (1143.2 ± 412.4 (pmol/g FW)); SA concentration was approximately one-third of this level. SA levels in stems were comparable to JA levels. Notably, BL was only detected in leaf extracts (4.6 ± 0.8 (pmol/g FW)). Hormone profiles have been determined to be diverse in tissues of *L. japonicus*.

## No significant difference of plant hormone content in *L. japonicus* between the two methods

We subsequently constructed a matrix standard curve by adding actual samples into standard solutions for quantification of phytohormones. Plant hormones were quantified in the same tissue extracts of *L. japonicus* as extracts used for stable isotope dilution (Table 3).

Leaf extract with the highest SA content (2480.9 ± 198.8 (pmol/g FW)) and the root extract with the highest JA concentration (1129.7 ± 347.4 (pmol/g FW)) were found. Phytohormone content was compared with the results from stable isotope dilution analysis (Fig 3A–3C).

**Table 1. Matrix effects in plant hormone determination.**

| Tissue | Compound | Matrix effect (%) | S.D. |
|--------|----------|-------------------|------|
| Leaf | IAA | 72.9 | 4.4 |
| | cZ | 37.2 | 9.6 |
| | ABA | 53.8 | 3.7 |
| | GA$_4$ | 51.1 | 3.7 |
| | SA | 54.5 | 14.7 |
| | JA | 50.9 | 15.8 |
| | BL | 67.9 | 4.7 |
| Root | IAA | 46.2 | 4.0 |
| | cZ | 10.2 | 4.6 |
| | ABA | 49.1 | 3.4 |
| | GA$_4$ | 66.9 | 2.7 |
| | SA | 37.4 | 12.0 |
| | JA | 59.8 | 13.9 |
| | BL | 87.3 | 1.6 |
| Stem | IAA | 62.9 | 9.4 |
| | cZ | 33.5 | 6.6 |
| | ABA | 42.9 | 7.6 |
| | GA$_4$ | 50.5 | 1.2 |
| | SA | 57.3 | 9.0 |
| | JA | 49.9 | 12.4 |
| | BL | 77.5 | 0.4 |

Hormones were extracted from the tissues of *Lotus japonicus* and mixed with seven pure standards. Ten ng/mL of phytohormone mixtures was analyzed with or without the matrix, and the peak areas were compared. Matrix effects are the mean values from three biological replicates.

**Table 2. A stable isotope-based quantification of plant hormones.**

| Tissue | Compound | Linear range (ng/ml) | Curve | $R^2$ | LOD ng/ml | LOQ ng/ml | Content pmol/g FW | SD | RSD (%) |
|---|---|---|---|---|---|---|---|---|---|
| Leaf | IAA | 1.0–250.0 | Y = 0.0479516 + 0.0808471*X | 0.9999 | 0.03 | 0.1 | 117.5 | 14.1 | 12.0 |
| | cZ | 1.0–250.0 | Y = −0.119872 + 0.0903252*X | 0.9988 | 0.1 | 0.2 | N.D. | N.D. | N.D. |
| | ABA | 1.0–250.0 | Y = −0.0962675 + 0.0481031*X | 0.9992 | 0.02 | 0.1 | 219.2 | 102.7 | 46.9 |
| | GA$_4$ | 1.0–250.0 | Y = −0.730001 + 0.102525*X | 0.9934 | 0.01 | 0.02 | N.D. | N.D. | N.D. |
| | SA | 1.0–250.0 | Y = 0.0388554 + 0.0687677*X | 0.9967 | 0.1 | 0.4 | 2397.5 | 203.5 | 8.5 |
| | JA | 1.0–250.0 | Y = −0.210579 + 0.101196*X | 0.9990 | 0.02 | 0.1 | 1042.9 | 113.6 | 10.9 |
| | BL | 1.0–250.0 | Y = −0.0247326 + 0.0846773*X | 0.9994 | 0.01 | 0.02 | 4.6 | 0.8 | 18.6 |
| Root | IAA | 1.0–250.0 | Y = 0.0479516 + 0.0808471*X | 0.9999 | 0.03 | 0.1 | 84.9 | 5.0 | 5.9 |
| | cZ | 1.0–250.0 | Y = −0.119872 + 0.0903252*X | 0.9988 | 0.1 | 0.2 | N.D. | N.D. | N.D. |
| | ABA | 1.0–250.0 | Y = −0.0962675 + 0.0481031*X | 0.9992 | 0.02 | 0.1 | 47.4 | 9.9 | 20.8 |
| | GA$_4$ | 1.0–250.0 | Y = −0.730001 + 0.102525*X | 0.9934 | 0.01 | 0.02 | N.D. | N.D. | N.D. |
| | SA | 1.0–250.0 | Y = 0.0388554 + 0.0687677*X | 0.9967 | 0.1 | 0.4 | 297.8 | 89.8 | 30.1 |
| | JA | 1.0–250.0 | Y = −0.210579 + 0.101196*X | 0.9990 | 0.01 | 0.04 | 1187.5 | 349.8 | 29.5 |
| | BL | 1.0–250.0 | Y = −0.0247326 + 0.0846773*X | 0.9994 | 0.01 | 0.02 | N.D. | N.D. | N.D. |
| Stem | IAA | 1.0–250.0 | Y = 0.0479516 + 0.0808471*X | 0.9999 | 0.03 | 0.1 | 216.7 | 33.8 | 15.6 |
| | cZ | 1.0–250.0 | Y = −0.119872 + 0.0903252*X | 0.9988 | 0.1 | 0.2 | N.D. | N.D. | N.D. |
| | ABA | 1.0–250.0 | Y = −0.0962675 + 0.0481031*X | 0.9992 | 0.02 | 0.1 | 178.3 | 32.7 | 18.3 |
| | GA$_4$ | 1.0–250.0 | Y = −0.730001 + 0.102525*X | 0.9934 | 0.01 | 0.02 | N.D. | N.D. | N.D. |
| | SA | 1.0–250.0 | Y = 0.0388554 + 0.0687677*X | 0.9967 | 0.1 | 0.4 | 1758.5 | 227.2 | 12.9 |
| | JA | 1.0–250.0 | Y = −0.210579 + 0.101196*X | 0.9990 | 0.01 | 0.04 | 1357.1 | 168.6 | 12.4 |
| | BL | 1.0–250.0 | Y = −0.0247326 + 0.0846773*X | 0.9994 | 0.01 | 0.02 | N.D. | N.D. | N.D. |

Seven phytohormones were extracted and analyzed by LC-MS/MS. Calibration curves were constructed with pure standards without matrix, and the peak areas were corrected based on the peak areas of stable isotope-labelled internal standards. Plant hormone extraction was performed in biological triplicate. $R^2$, correlation coefficient; LOD, limit of detection; LOQ, limit of quantification; RSD, relative standard deviation; FW, fresh weight.

No significant difference was observed in the concentrations (pmol/g FW) between the two quantification methods (t-test, $p > 0.05$), and phytohormone profiles have exhibited similar patterns. We also evaluated repeatability (accuracy and precision) for quantification of plant hormones by standard addition method (S2 Table). Precision and accuracy, expressed as relative standard deviation (RSD) and relative error ranged from 3.5% to 16.3%, from −18.2% to +3.1%, respectively. The standard addition method is effective for quantification of hormones by correcting for matrix effects; thus it is applicable for the three major tissues of *L. japonicus*.

## Discussion

Phytohormones are essential signaling molecules in multiple physiological processes, including growth, development, and stress response. *L. japonicus* is a model legume widely used in studies on nitrogen-fixing symbiosis and arbuscular mycorrhizal (AM) symbiosis; however, plant hormone profiles remain to be fully elucidated to date. Only one report was found that measured gibberellins, including active GA$_1$, JA, SA, IAA, and ABA, which were detected in the roots. Further, GA$_1$ and its intermediates GA$_8$, GA$_{19}$, and GA$_{53}$ were significantly accumulated in response to arbuscular mycorrhizal-fungal infection [23]. We also examined the levels of the seven major plant hormones in the roots, stems, and leaves and found, for the first time, that plant hormones in stems and leaves contain high concentrations of salicylic and jasmonic acids. ABA and IAA were considerably higher in leaves and stems than in roots, and SA was

**Table 3. Quantification of plant hormones using standard addition.**

| Tissue | Compound | Linear range (ng/ml) | Curve | $R^2$ | LOD | LOQ | Content | | RSD (%) |
|---|---|---|---|---|---|---|---|---|---|
| | | | | | ng/ml | ng/ml | pmol/g FW | SD | |
| Leaf | IAA | 1.0–250.0 | Y = 77271*X − 141437 | 0.9995 | 0.1 | 0.5 | 95.1 | 4.0 | 4.3 |
| | cZ | 10.0–250.0 | Y = 35908*X + 162509 | 0.9963 | 0.5 | 1.6 | N.D. | N.D. | N.D. |
| | ABA | 1.0–250.0 | Y = 66926*X − 29050 | 0.9998 | 0.1 | 0.3 | 185.7 | 84.4 | 45.4 |
| | GA$_4$ | 1.0–250.0 | Y = 151474*X − 147592 | 0.9998 | 0.1 | 0.2 | N.D. | N.D. | N.D. |
| | SA | 1.0–250.0 | Y = 96955*X + 2118049 | 0.9917 | 0.2 | 0.7 | 2480.9 | 198.8 | 8.0 |
| | JA | 1.0–250.0 | Y = 69832*X − 1415498 | 0.9990 | 0.1 | 0.2 | 1023.8 | 94.2 | 9.2 |
| | BL | 1.0–250.0 | Y = 152721*X − 327260 | 0.9995 | 0.1 | 0.3 | 10.9 | 0.04 | 0.4 |
| Root | IAA | 1.0–250.0 | Y = 140337*X − 13011 | 0.9993 | 0.1 | 0.3 | 71.8 | 10.3 | 14.3 |
| | cZ | 10.0–250.0 | Y = 33924*X − 53632 | 0.9975 | 0.3 | 0.9 | N.D. | N.D. | N.D. |
| | ABA | 1.0–250.0 | Y = 92423*X − 229613 | 0.9995 | 0.1 | 0.3 | 38.8 | 2.8 | 7.1 |
| | GA$_4$ | 1.0–250.0 | Y = 170129*X − 524708 | 0.9989 | 0.02 | 0.1 | N.D. | N.D. | N.D. |
| | SA | 1.0–250.0 | Y = 175984*X + 796822 | 0.9954 | 0.1 | 0.4 | 438.2 | 113.9 | 26.0 |
| | JA | 1.0–250.0 | Y = 74179*X + 1697965 | 0.9977 | 0.02 | 0.1 | 1129.7 | 347.4 | 30.7 |
| | BL | 1.0–250.0 | Y = 116960*X − 543073 | 0.9985 | 0.1 | 0.4 | N.D. | N.D. | N.D. |
| Stem | IAA | 1.0–250.0 | Y = 103600*X − 263583 | 0.9987 | 0.1 | 0.2 | 239.6 | 11.0 | 4.6 |
| | cZ | 10.0–250.0 | Y = 33302*X + 279101 | 0.9849 | 0.4 | 1.4 | N.D. | N.D. | N.D. |
| | ABA | 1.0–250.0 | Y = 78533*X − 134102 | 0.9986 | 0.1 | 0.4 | 161.4 | 20.2 | 12.5 |
| | GA$_4$ | 1.0–250.0 | Y = 199797*X − 732980 | 0.9986 | 0.02 | 0.1 | N.D. | N.D. | N.D. |
| | SA | 1.0–250.0 | Y = 114085*X − 1541232 | 0.9866 | 0.5 | 1.6 | 1617.9 | 351.3 | 21.7 |
| | JA | 1.0–250.0 | Y = 57977*X + 2093537 | 0.9962 | 0.1 | 0.2 | 1430.1 | 462.6 | 32.3 |
| | BL | 1.0–250.0 | Y = 132286*X − 539659 | 0.9973 | 0.1 | 0.2 | N.D. | N.D. | N.D. |

Seven phytohormones were extracted, and the calibration curves were constructed with sample matrix and analyzed by LC-MS/MS. Plant hormone extraction was performed in biological triplicate. $R^2$, correlation coefficient; LOD, limit of detection; LOQ, limit of quantification; RSD, relative standard deviation; FW, fresh weight.

higher in the root than the other parts, suggesting the existence of tissue-specific plant hormone regulation in *L. japonicus*. Plant hormones are fundamental signaling molecules that respond to biotic and abiotic stress [1]. Indeed, CV was comparatively large in this present study, implying that individual differences in hormones were evident and might fluctuate in response to external stimuli. Further studies would connect plant hormone profiles with plant hormone synthesis/transport network, and it will identify underlying mechanisms of hormonal crosstalk in host-bacterium mutualism. Unveiling plant hormone profiles in model legume, *L. japonicus*, might allow the enhancement of yields of legume crops, such as soybean since plant hormones have been implicated in plant growth and seed yield in legume [24].

Matrix effects, including ion enhancement or suppression, can often hamper accurate quantification of target analytes by LC-ESI-MS/MS. Mitigation efforts, such as purification of samples and applying proper calibration methods, are currently in use. Typically, plant hormones are purified along with spiked stable isotopes to correct for matrix effects [7]. However, stable isotope-labelled compounds for specific plant hormones are expensive and sometimes not commercially available, which limits their use. Although standard addition method is relatively more complex from a methodological point of view compared to stable isotope dilution, it is an effective method since it requires no stable isotope-labelled targets; this method is employed to detect drugs and hormones in plasma and sewage sludge [12,20,21] and pesticides in crop and feedstock [25].

We validated the standard addition method to quantify seven major plant hormones and measured their levels in the tissues of *Lotus japonicus*. This species is a legume, a family of

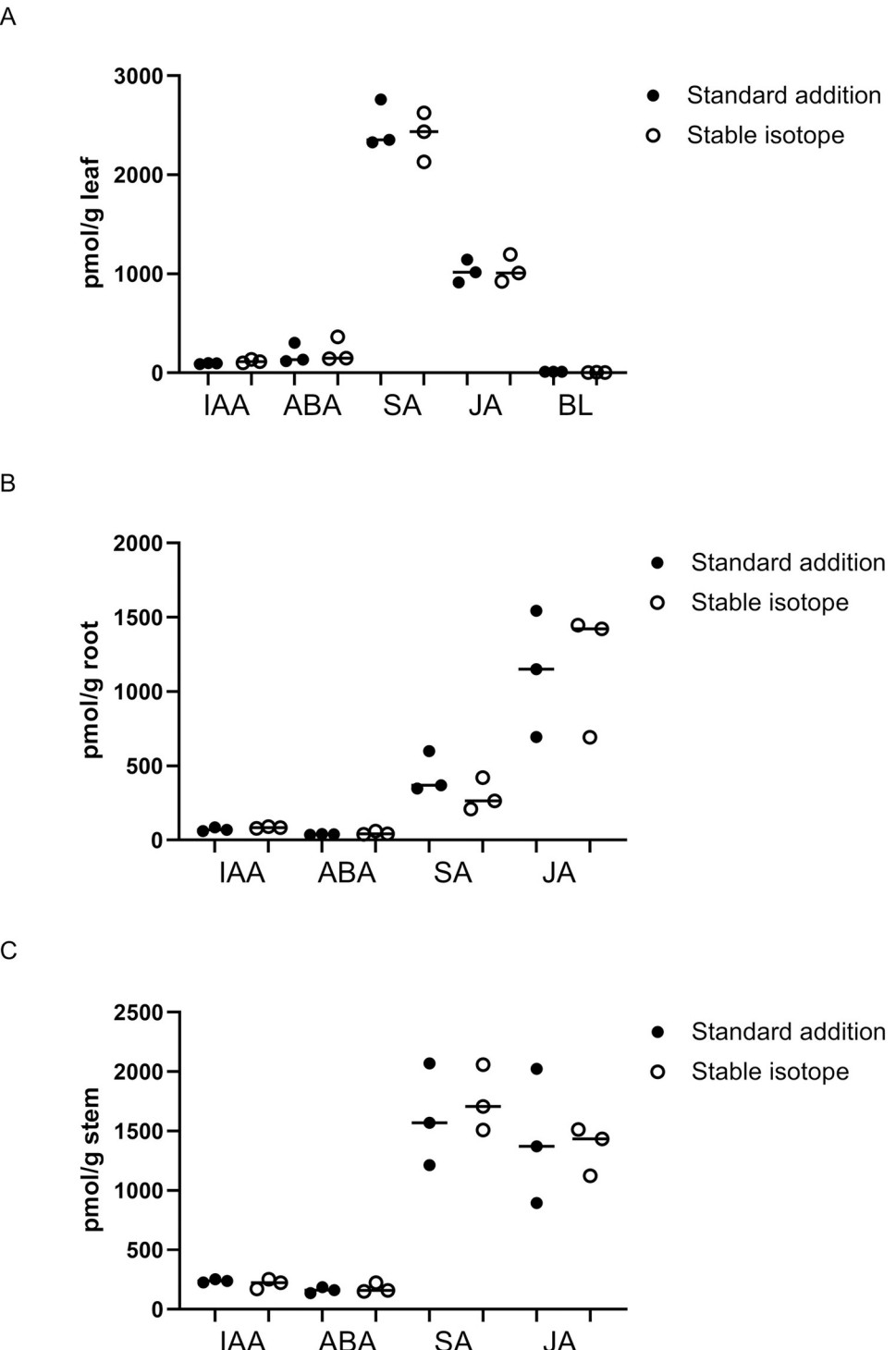

**Fig 3. Comparison of standard addition method with stable isotope dilution method.** Plant hormone contents in the leaves (A), roots (B), and stems (C) were compared between standard addition and stable isotope addition methods. Significant differences were not observed.

principal crops like soybeans. Understanding plant hormone fluctuation in *Lotus japonicus* is of great interest in plant physiology [26]. The standard addition method requires a sample matrix with several concentrations of standard solution. This method is thought to be inappropriate for low concentration samples. However, recent developments in detecting the sensitivity of mass spectrometry have largely solved this problem, and a small quantity of sample (20–50 mg fresh weight per standard level) can now be enough to quantify phytohormones at attomole levels [6]. Our data using high-resolution mass spectrometry (HR-MS) detected femtomolar concentrations of plant hormones, a sensitivity lower than previous study using triple quadrupole mass spectrometry and ultra high performance liquid chromatography (UHPLC) column (particle size of 1.7 μm). Sensitivity might be improved if a UHPLC column is used. Selective reaction monitoring (SRM) has been widely used [7], but selected ion monitoring using HR-MS [27,28] has never been applied to simultaneous quantification of plant hormones. Stable isotope labelling has been considered to be ideal for matrix correction, and our data show that this method displays lower LOD, LOQ, and CV than standard addition. However, standard addition would be beneficial when corresponding stable isotopes are unavailable. Further, standard addition would also be applicable to other parts of the *L. japonicus*, such as nodule, flower, and pods, and likely other plant species.

Gibberellin $A_3$ is a well-known active plant hormone, but a stable isotope-labelled form is not commercially available. Stable isotope labelling often involves culturing cells or organisms in a medium that contains stable isotope ($^2$H-, $^{13}$C-, or $^{15}$N-)-labelled molecule building blocks [29]. Another method is to synthesize a precursor of the target compound and then use stable isotope-labelled substrate for the final reaction step [30,31]. In either case, stable isotope labelling often requires time, expense, and multiple purification steps. These issues hinder synthesis in many laboratories. Purification of plant hormones requires large amounts of organism grown in medium containing stable isotopes, since plant hormones exist in plants in trace amounts. Organic chemical synthesis for plant hormones also requires multiple reaction and purification steps in order to obtain a final product. We then validated and quantified plant hormone content in *L. japonicus* using a standard addition method without stable isotopes. The method was compared with stable isotope dilution, and similar plant hormone profiles in three plant organs were obtained. Next, the method would be evaluated using *L. japonicus* under hormone-inducing stresses such as drought or pathogen infection as well as Lotus retrotransposon 1 (LORE1) mutants which have mutations in plant hormone biosynthetic pathways. Since plant hormones in distinct organs can critically affect phenotypes, such as growth and differentiation, by plant hormone crosstalk [2], our method requiring no stable isotopes will facilitate understanding of plant hormonomics in the future.

## Supporting information

**S1 Fig. Preparation of samples using standard addition.** The samples after solid phase extraction (SPE) were mixed with standard solution to construct matrix calibration curve ranging from 1.0 ng/ml to 250.0 ng/ml. The target sample was diluted with 50% methanol in a dilution factor 1.05 (50 μl/47.5 μl). All samples were analyzed by LC-MS/MS analysis. (TIF)

**S2 Fig. Representative SIM chromatogram of seven phytohormones.** One hundred ng/ml of seven phytohormone mixture was analyzed in Quadrupole-orbitrap mass spectrometry. (TIF)

**S3 Fig. Representative SIM chromatogram of stable isotope-labeled seven phytohormones.** Ten ng/ml of stable isotope-labeled phytohormones was analyzed in Quadrupole-orbitrap

mass spectrometry.
(TIF)

**S1 Table. Targeted-selected ion monitoring (t-SIM) mode to detect seven plant hormones in Quadrupole-orbitrap mass spectrometer.**
(TIF)

**S2 Table. Recovery rate and repeatability to validate standard addition method with the tissues of *Lotus japonicus*.** The tissue extracts with 100 ng/ml phytohormones were subjected to solid phase extraction (SPE) and the peak areas of an analyte were determined by LC-MS/MS analysis. The samples spiked with phytohormones after SPE were also analyzed. For repeatability, the tissue extracts with 125 ng/ml phytohormones were also subjected to SPE and quantification by standard addition method. The experiments were repeated three times and the mean of recovery rate, relative standard deviation (RSD) and relative error were calculated.
(TIF)

## Author Contributions

**Data curation:** Takuyu Hashiguchi.

**Investigation:** Takuyu Hashiguchi, Koki Fukushima.

**Writing – original draft:** Takuyu Hashiguchi.

**Writing – review & editing:** Masatsugu Hashiguchi, Hidenori Tanaka, Takahiro Gondo, Ryo Akashi.

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
