## [Decision Letter · Decision Letter 0]

30 Nov 2020

PONE-D-20-34002

Quantitative analysis of the seven plant hormones in Lotus japonicus using standard addition method

PLOS ONE

Dear Dr. Akashi,

Thank you for submitting your manuscript to PLOS ONE. After careful consideration, we feel that it has merit but does not fully meet PLOS ONE’s publication criteria as it currently stands. Therefore, we invite you to submit a revised version of the manuscript that addresses the points raised during the review process.

The manuscript has been revised by two reviewers that appreciated it. However, some comments should be addressed before the publication.

We look forward to receiving your revised manuscript.

Kind regards,

Raffaella Balestrini

Academic Editor

PLOS ONE

Journal Requirements:

2. During your revisions, please note that a simple title correction is required: to follow correct English language usage, the title should read "Quantitative analysis of seven plant hormones in Lotus japonicus using standard addition method". Please ensure this is updated in the manuscript file and the online submission information.

2This work was supported by Toyota Motor Corporation and the National BioResource

Project (NBRP) of the Japan Agency for Medical Research and Development (AMED)."

Additionally, because some of your funding information pertains to commercial funding, we ask you to provide an updated Competing Interests statement, declaring all sources of commercial funding.

In your Competing Interests statement, please confirm that your commercial funding does not alter your adherence to PLOS ONE Editorial policies and criteria by including the following statement: "This does not alter our adherence to PLOS ONE policies on sharing data and materials.” as detailed online in our guide for authors  http://journals.plos.org/plosone/s/competing-interests.  If this statement is not true and your adherence to PLOS policies on sharing data and materials is altered, please explain how.

Please include the updated Competing Interests Statement and Funding Statement in your cover letter. We will change the online submission form on your behalf.

Reviewers' comments:

Reviewer's Responses to Questions

**Comments to the Author**

1. Is the manuscript technically sound, and do the data support the conclusions?

Reviewer #1: Yes

Reviewer #2: Yes

2. Has the statistical analysis been performed appropriately and rigorously? 

Reviewer #1: Yes

Reviewer #2: Yes

3. Have the authors made all data underlying the findings in their manuscript fully available?

Reviewer #1: Yes

Reviewer #2: Yes

4. Is the manuscript presented in an intelligible fashion and written in standard English?

Reviewer #1: Yes

Reviewer #2: Yes

5. Review Comments to the Author

Reviewer #1: In this work, Hashiguchi and colleagues present a methodological manuscript in which they describe the quantitative analysis of seven plant hormones in Lotus japonicus roots, shoots and stems, with high-resolution mass spectrometry (HR-MS).

They showed an important technical analysis about the impact of matrix effect in the purification and quantification of hormones that can vary between 10 and 87%, therefore ion suppression can largely decrease the peak area of phytohormones. And among the main results, they demonstrated that there were no significant differences between standard addition and stable isotope addition methods, therefore allowing to quantify hormones without any treatment with stable isotopes.

Altogether the work is technically sound and the approaches are of wide interest for the community.

A further validation of the methods on a perturbed system would add some extra value to the work. For example the Authors could challenge the plants with one relevant hormone-inducing stress (just as an example drought stress to induce ABA) and further compare ABA concentration with a physiological condition. Another option would be to test a Lotus japonicus known mutant for any hormonal biosynthetic pathway.

In addition to this, I have a few specific comments:

Figure 3: Please do not use bar graphs but show all the measurements (refer to Weissgerber et al., 2015, Plos Biology, https://doi.org/10.1371/journal.pbio.1002128): this would

Finally to better validate the method, a calculation of repeatability (accuracy and precision) obtained during the validation of the method could constitute an added value. By repeating the same measurements on the same starting material how much variance are you observing? Such as done in the cited work by Trapp et al., Frontiers in Plant Science 2014. I think this is particularly important if the Authors are not adding any experiments on Lotus mutants and/or stress conditions that are inducing different hormonal changes.

As a suggestion, to improve the readability, consider to change the paragraph title by stating the key results obtained in that paragraph. For example instead of: “Quantification of plant hormones by standard addition and comparison of the two quantification methods” you could state that there is no significant difference between the two methods.

line 45-46: There are more than 8 groups of hormones: please, at least, also consider strigolactones

line 340-343: repetition of a full sentence, please correct it.

Reviewer #2: The paper “Quantitative analysis of the seven plant hormones in Lotus japonicus using standard addition method” is an interesting methodological manuscript which aims to compare standard addition and stable isotope dilution method to quantify plant hormones in different plant tissues. The authors have utilized leaves, roots and stems of Lotus japonica to test if the standard addition method could be a robust and accurate method to quantify plant hormones in plant tissues. The work was conducted utilizing a rigorous methodology and furnish a validated method to quantify plant hormones, in particular when it is difficult to find the appropriate internal standard. However, I have two main comments:

- the method is more complex, from a methodological point of view respect to the method of internal standard. I think this should be mentioned in the discussion (in addition to highlight the strong points of this method);

- the discussion should be revised in a more logical way, starting from the role of hormones and their importance in AMF- and bacterium-plant interaction (line 314-329) and then moving to the discussion of the results presented in the paper (validation of the standard addition method and comparison with the internal standard method) (line 295-313 and 330-345).

I suggest to revise also some minor points:

-line 26 “THEIR content levels vary depending on the species, and THEY also change...” I think this sentence is referred to plant hormones.

-line 55 “ionizing hydrophobic analytes”..not all plant hormones are hydrophobic substances (ex. salicylic acid, abscisic acid)..I suggest to revise this point

-line 107 “Extraction for plant hormones has BEEN CONDUCTED UTILIZING...”

-line 136: the authors have already described that extracts were purified using SPE. Here it can be said “Purified extracts were separated ....”

- line 328-329: the last sentence of the paragraph is very speculative. I suggest to revise this sentence.

6. PLOS authors have the option to publish the peer review history of their article (what does this mean?). If published, this will include your full peer review and any attached files.

Reviewer #1: No

Reviewer #2: **Yes: **Cecilia Brunetti

---

## [Author Response · Author response to Decision Letter 0]

23 Dec 2020

Response to Reviewer 1

We express our sincere gratitude to this reviewer for the questions, comments, and suggestions provided to us about our manuscript. In the revised manuscript, we have taken into account the suggestions of the reviewer and made the necessary revisions accordingly. Our answers to your questions, comments, and suggestions are listed below following each specific question/comment.

Comment 1: In this work, Hashiguchi and colleagues present a methodological manuscript in which they describe the quantitative analysis of seven plant hormones in Lotus japonicus roots, shoots and stems, with high-resolution mass spectrometry (HR-MS). They showed an important technical analysis about the impact of matrix effect in the purification and quantification of hormones that can vary between 10 and 87%, therefore ion suppression can largely decrease the peak area of phytohormones. And among the main results, they demonstrated that there were no significant differences between standard addition and stable isotope addition methods, therefore allowing to quantify hormones without any treatment with stable isotopes. Altogether the work is technically sound and the approaches are of wide interest for the community. A further validation of the methods on a perturbed system would add some extra value to the work. For example the Authors could challenge the plants with one relevant hormone-inducing stress (just as an example drought stress to induce ABA) and further compare ABA concentration with a physiological condition. Another option would be to test a Lotus japonicus known mutant for any hormonal biosynthetic pathway.

Our response: We have added the importance of the works you suggested in lines 351-354 in the discussion. 

Comment 2: Figure 3: Please do not use bar graphs but show all the measurements (refer to Weissgerber et al., 2015, Plos Biology, https://doi.org/10.1371/journal.pbio.1002128): this would

Our response: We have replaced the bar graphs in Figure 3 with the scatter plots as you suggested.

Comment 3: Finally to better validate the method, a calculation of repeatability (accuracy and precision) obtained during the validation of the method could constitute an added value. By repeating the same measurements on the same starting material how much variance are you observing? Such as done in the cited work by Trapp et al., Frontiers in Plant Science 2014. I think this is particularly important if the Authors are not adding any experiments on Lotus mutants and/or stress conditions that are inducing different hormonal changes.

Our response: We have added the data on repeatability (accuracy and precision) by standard addition method in supplemental Table 2 and in lines 284-287. Accordingly, we have added the sentences in lines 464-465 and 467-470 in the table legend. 

Comment 4: As a suggestion, to improve the readability, consider to change the paragraph title by stating the key results obtained in that paragraph. For example instead of: “Quantification of plant hormones by standard addition and comparison of the two quantification methods” you could state that there is no significant difference between the two methods.

Our response: We have changed the paragraph title to “No significant difference of plant hormone content in L. japonicus between the two methods” in lines 255-256 and “Non-negligible matrix effects of root, stem, and leaf from Lotus japonicus in plant hormone determination” in lines 193-194.

Comment 5: line 45-46: There are more than 8 groups of hormones: please, at least, also consider strigolactones

Our response: We have included “strigolactones” in line 46 and changed from “eight” to “nine” in line 45. 

Comment 6: line 340-343: repetition of a full sentence, please correct it.

Our response: We have deleted the repetitive sentence “Thus, the standard addition method was comparable to the method using stable isotope labelling.” in this revised manuscript.

Response to reviewer 2

We express our sincere gratitude to this reviewer for the questions, comments, and suggestions provided to us about our manuscript. In the revised manuscript, we have taken into account the suggestions of the reviewer and made the necessary revisions accordingly. Our answers to your questions, comments, and suggestions are listed below following each specific question/comment.

Comment 1: The paper “Quantitative analysis of the seven plant hormones in Lotus japonicus using standard addition method” is an interesting methodological manuscript which aims to compare standard addition and stable isotope dilution method to quantify plant hormones in different plant tissues. The authors have utilized leaves, roots and stems of Lotus japonica to test if the standard addition method could be a robust and accurate method to quantify plant hormones in plant tissues. The work was conducted utilizing a rigorous methodology and furnish a validated method to quantify plant hormones, in particular when it is difficult to find the appropriate internal standard. However, I have two main comments:

- the method is more complex, from a methodological point of view respect to the method of internal standard. I think this should be mentioned in the discussion (in addition to highlight the strong points of this method);

Our response: We have added “Although standard addition method is relatively more complex from a methodological point of view compared to stable isotope dilution,” in lines 315-316 in the discussion. 

Comment 2: - the discussion should be revised in a more logical way, starting from the role of hormones and their importance in AMF- and bacterium-plant interaction (line 314-329) and then moving to the discussion of the results presented in the paper (validation of the standard addition method and comparison with the internal standard method) (line 295-313 and 330-345).

Our response: As you suggested, we have started from the role of hormones in lines 291-293 and replaced the second paragraph (lines 295-313) with the first paragraph (lines 314-329) to discuss the results more logically.

Comment 3: -line 26 “THEIR content levels vary depending on the species, and THEY also change...” I think this sentence is referred to plant hormones.

Our response: We have changed the words as you pointed out in line 26 and 27.

Comment 4: -line 55 “ionizing hydrophobic analytes”..not all plant hormones are hydrophobic substances (ex. salicylic acid, abscisic acid)..I suggest to revise this point

Our response: We have revised as you suggested in line 55.

Comment 5: -line 107 “Extraction for plant hormones has BEEN CONDUCTED UTILIZING...”

Our response: We have revised as you suggested in line 109.

Comment 6: -line 136: the authors have already described that extracts were purified using SPE. Here it can be said “Purified extracts were separated ....”

Our response: We have revised as you suggested in line 139.

Comment 7: - line 328-329: the last sentence of the paragraph is very speculative. I suggest to revise this sentence.

Our response: We have changed the sentence to “Unveiling plant hormone profiles in model legume, L. japonicus, might allow the enhancement of yields of legume crops, such as soybean since plant hormones have been implicated in plant growth and seed yield in legume [24].” in lines 307-309. We also added a new reference “Wilkinson S, Kudoyarova GR, Veselov DS, Arkhipova TN, Davies WJ. Plant hormone interactions: innovative targets for crop breeding and management. J Exp Bot. 2012;63: 3499-3509.” in line 309. Accordingly, the references 25-31 have been reordered.

---

## [Decision Letter · Decision Letter 1]

4 Feb 2021

Quantitative analysis of seven plant hormones in Lotus japonicus using standard addition method

PONE-D-20-34002R1

Dear Dr. Akashi,

We’re pleased to inform you that your manuscript has been judged scientifically suitable for publication and will be formally accepted for publication once it meets all outstanding technical requirements.

Kind regards,

Raffaella Balestrini

Academic Editor

PLOS ONE

Additional Editor Comments (optional):

Reviewers' comments:

Reviewer's Responses to Questions

**Comments to the Author**

1. If the authors have adequately addressed your comments raised in a previous round of review and you feel that this manuscript is now acceptable for publication, you may indicate that here to bypass the “Comments to the Author” section, enter your conflict of interest statement in the “Confidential to Editor” section, and submit your "Accept" recommendation.

Reviewer #1: All comments have been addressed

Reviewer #2: All comments have been addressed

2. Is the manuscript technically sound, and do the data support the conclusions?

Reviewer #1: Yes

Reviewer #2: Yes

3. Has the statistical analysis been performed appropriately and rigorously? 

Reviewer #1: Yes

Reviewer #2: Yes

4. Have the authors made all data underlying the findings in their manuscript fully available?

Reviewer #1: Yes

Reviewer #2: Yes

5. Is the manuscript presented in an intelligible fashion and written in standard English?

Reviewer #1: Yes

Reviewer #2: Yes

6. Review Comments to the Author

Reviewer #1: Please consider to correct line 351. You wrote: "the method would be evaluated using L. japonicus" but it could be better to write that it will be important to validate the method with using...

Reviewer #2: In this revised version of the paper, the authors have addressed all the concerns and suggestions. I am satisfied by all changes carried out and endorse the publication of the manuscript.

7. PLOS authors have the option to publish the peer review history of their article (what does this mean?). If published, this will include your full peer review and any attached files.

Reviewer #1: **Yes: **Marco Giovannetti

Reviewer #2: **Yes: **Cecilia Brunetti

---

## [Editor Report · Acceptance letter]

8 Feb 2021

PONE-D-20-34002R1 

Quantitative analysis of seven plant hormones in *Lotus japonicus* using standard addition method 

Dear Dr. Akashi:

I'm pleased to inform you that your manuscript has been deemed suitable for publication in PLOS ONE. Congratulations! Your manuscript is now with our production department. 

Kind regards, 

on behalf of

Dr Raffaella Balestrini 

Academic Editor

PLOS ONE